# Mesenteric Ischemia in a Patient with Essential Thrombocythemia: Does COVID-19 Play Any Role? A Case Report and Overview of the Literature

**DOI:** 10.3390/medicina58091147

**Published:** 2022-08-24

**Authors:** Mihail Cotorogea-Simion, Sebastian Isac, Alina Tita, Letitia Toma, Laura Elena Iliescu, Adriana Mercan-Stanciu, Teodora Isac, Anca Bobirca, Florin Bobirca, Cristian Cobilinschi, Maria Daniela Tanasescu, Gabriela Droc

**Affiliations:** 1Department of Anesthesiology and Intensive Care I, ‘Fundeni’ Clinical Institute, 022328 Bucharest, Romania; 2Department of Physiology, Faculty of Medicine, Carol Davila University of Medicine and Pharmacy, 020021 Bucharest, Romania; 3Department of Internal Medicine II, Fundeni Clinical Institute, 022328 Bucharest, Romania; 4Department of Rheumatology, Dr. Ion Cantacuzino Hospital, 073206 Bucharest, Romania; 5Department of General Surgery, Dr. Ion Cantacuzino Hospital, 073206 Bucharest, Romania; 6Department of Anesthesiology and Intensive Care, Clinical Emergency Hospital, 014461 Bucharest, Romania; 7Department of Medical Semiology, Discipline of Internal Medicine I and Nephrology, Faculty of Medicine, Carol Davila University of Medicine and Pharmacy, 020021 Bucharest, Romania

**Keywords:** mesenteric ischemia, essential thrombocythemia, COVID-19 coagulopathy, endovascular therapy, atherosclerosis

## Abstract

Introduction: Chronic mesenteric ischemia is a rare entity with non-specific symptomatology; combined with rare etiologies, it could lead to unwarranted surgical indication. Case report: We report the case of an 85-year-old woman, with a history of hypertension, persistent thrombocytosis, atherosclerosis, and recent minor COVID-19 infection, presenting to the hospital with postprandial abdominal pain and nonspecific clinical examination findings; upon abdominal CT, superior mesenteric artery circumferential thrombosis was revealed. A bone marrow biopsy was performed due to suspected essential thrombocythemia, confirming the diagnosis. An endovascular approach was chosen as therapy option and a stent was placed in the occluded area. Dual antiplatelet and cytoreductive therapies were initiated after the intervention. Clinical course was excellent, with no residual stenosis 1 month after stenting. Conclusions: The therapeutic strategy in elderly patients with exacerbated chronic mesenteric ischemia requires an interdisciplinary approach in solving both the exacerbation and the underlying conditions in order to prevent further thrombotic events. Although the patient presented a thrombotic state, other specific risk factors such as COVID-19 related-coagulopathy and essential thrombocythemia should be considered.

## 1. Introduction

Mesenteric ischemia (MI) is a clinical entity which arises when the blood supply fails to meet the metabolic demands of the small intestine. This can occur either through arterial (superior mesenteric and celiac artery) or venous (mesenteric veins) flow obstruction, or through non-obstructive means [1,2,3].

Based on symptom severity and duration until presentation, MI can be divided into acute, subacute, and chronic; chronic mesenteric ischemia has a prevalence of 1:100,000, whereas acute MI is the cause of 0.1% of hospital admissions [1]. The most commonly incriminated mechanism for acute MI is mesenteric artery embolism, accounting for around half of the cases, followed by thrombosis on an atherosclerotic arterial wall, in approximately a quarter of the patients, then mesenteric venous system occlusion by thrombi (5–15% of all acute MI cases), and non-occlusive mechanisms (roughly the same proportion as the previous) [1,2]. The arterial emboli might originate from the heart (as is the case with atrial fibrillation, endocarditis, or cardiomyopathy) or from a ruptured aortic plaque [1,2]. The superior mesenteric artery’s acute branching angle from the aorta and its large diameter facilitate entrance for upstream emboli; however, the same large diameter means that, usually, the proximal branches are spared in the case of an embolism, sparing the duodenal, proximal jejunal, and colonic blood supply [1,2]. Arterial thrombosis occurs most frequently in association with atherosclerosis, but can also be triggered by vasculitis, dissection, or aneurysms [2]. Mesenteric venous thrombosis is associated with stasis, as one would encounter in liver cirrhosis, abdominal organ inflammation (pancreatitis, inflammatory bowel disorders, appendicitis) or trauma, and with thrombophilia (inherited defects in the coagulation cascade—protein C, S, antithrombin deficiency, factor V Leiden mutation; acquired disorders—thrombocythemia, antiphospholipid syndrome, nephrotic syndrome; iatrogenic—oral contraceptives) [4]. Nonocclusive mesenteric ischemia is a result of intestinal hypoperfusion, which can be caused by global circulatory failure, as might be encountered in cardiac disease, hemodialysis-induced hypotension, or shock of various etiologies—hemorrhagic, septic, hypovolemic, etc., or by local factors [3]. Mesenteric vasospasm diverts blood flow away from the bowels, causing acute MI. Vasospasms are mainly drug-induced: digoxin, phenylephrine, vasopressin, calcium channel blockers, amphetamines, or cocaine [3]. Other situations when blood flow is routed away from the intestines, leading to ischemia, include increased intraabdominal pressure (extrinsic compression of intestinal vascular supply) and extreme physical effort (splanchnic vasoconstriction, blood flowing mainly towards the muscles) [3]. Aggressive enteral feeding is another condition that could precipitate an acute MI episode, as it increases metabolic demand in an already hypoperfused area (it usually occurs in ICU patients, who are more often in shock states than the general population) [3]. 

The patient with acute MI usually presents severe abdominal pain, which appears unjustified by the clinical examination findings (soft abdomen, no pain upon palpation, intense borborygmi upon auscultation), accompanied by nausea and vomiting; once tissue necrosis sets in, the patient develops fever, shock, rebound tenderness, abdominal wall rigidity, and guarding [1]. The intestinal sounds become absent, suggesting a lack of peristalsis; this is nevertheless associated with melena or hematochezia [1].

Subacute MI is most often caused by mesenteric venous thrombosis, since intestinal oxygen supply gradually decreases, as the veins become engorged and blood pools up in the bowel wall; however, infarction is not necessarily the logical conclusion, as collaterals can form, given enough time [4,5,6].

The clinical presentation of subacute MI is similar to acute MI, but the onset is slightly more insidious, with initial symptoms being constipation or nausea, which causes patients to delay presentation to the emergency department (reported mean duration—6 to 14 days) [4,7]. Patients suffering from subacute MI are significantly younger than other mesenteric ischemia patients, usually 40 to 60 years old, with 75% of the cases occurring in males [7].

Chronic MI is caused by atherosclerosis in an overwhelming majority of cases (>90%); other causes are large vessel vasculitis (such as Takayasu arteritis), radiation-induced vascular injury, median arcuate ligament syndrome (cyclical celiac trunk compression against the diaphragmatic crura due to aberrant median arcuate ligament positioning, which ameliorates during inspiration), and mesenteric venous system thrombosis [2,8].

Chronic MI (CMI) manifests itself as a history of abdominal pain after meals and weight loss (patients learn to avoid the pain by eating smaller meals). This occurs secondary to long-standing obstruction in superior mesenteric artery flow. The bowels are able to develop collateral circulation from the pancreaticoduodenal artery, a branch of the celiac trunk, which can supply enough blood during a resting state, but can’t account for acute increases in oxygen demand (such as after a particularly fatty meal, when blood flow to the intestines can increase by almost 100%) [1,2]. The typical CMI is female (the male-to-female ratio is reversed when compared to subacute MI), older than 60 years of age [1].

Even if CMI occurs mostly in the elderly, no correlation was found between the presence of mesenteric stenosis and death at 6.5 years in a multicentric, prospective study involving 553 patients with no abdominal complaints [9]. Conversely, older patients with histologically proven ischemic colitis seemed to have an increased risk of death if the following risk factors were present: constipation, hepatitis C virus infection, cancer, vasculopathy, ascending colon involvement, increased blood urea or lactate dehydrogenases [10]. Moreover, episodes of exacerbation could also appear regardless of the dietary habits of the patients, which requires urgent diagnostic and therapy.

Concurrent hematologic diseases or post-COVID syndrome are rare conditions that could also exacerbate mesenteric angina in patients suffering from extensive atherosclerosis.

COVID-19 infection could also predispose to mesenteric angina episodes, especially in patients with advanced atherosclerosis. A recent study published by Xie et al. demonstrated that COVID-19 increases the incidence of cardiovascular disease, including ischemia and venous thromboembolism, at 1 year, regardless of the severity of the disease [11]. Although it is difficult to assess the exact causality, the main mechanism proposed for these ischemic events involves endothelial inflammation secondary to angiotensin-converting enzyme receptor (ACE-R) dysregulation combined with platelet dysfunction [12,13,14]. Moreover, patients with preexisting atherosclerosis are prone to develop ischemic events [13,14].

Essential thrombocythemia (ET) is a chronic Philadelphia-negative myeloproliferative neoplasm, whose main clinical manifestation is represented by thrombosis. The diagnostic criteria include platelet count over 450,000 mm^3^, suggestive bone marrow biopsy, exclusion of other myelodysplastic syndromes, and the absence of reactive thrombocytosis [15]. An important feature of the disease is represented by the JAK-2 mutation status. Patients with mutant JAK-2 are usually older and are more likely to develop thrombosis [16].

Our study reports a rare case of exacerbated chronic mesenteric ischemia in a patient with newly diagnosed JAK-2-mutated ET after a minor form of COVID-19 infection. The aim of this case report is to raise r clinician awareness regarding thrombotic states that could aggravate chronic mesenteric ischemia, and highlighting the need for minimally invasive (i.e., interventional) treatment in such patients.

## 2. Case Presentation

An 85-year-old, 68 kg woman was admitted to our clinic with a long history of coronary heart disease, hypertension, carotid atherosclerosis (Figure 1), and persistent thrombocytosis. Additionally, the patient had suffered from a minor form of COVID-19 infection 3 months prior, presenting mostly upper respiratory tract symptoms—cough, anosmia, and changes in taste, for which she received symptomatic therapy in ambulatory settings [17].

At clinical presentation, the patient reported severe postprandial pain in the lower abdomen, left flank, and left hypochondrium. The pain had appeared a week earlier and progressively worsened, especially after meals. She also described vomiting and sitophobia, but without changes in intestinal transit or any other symptoms. On admission, her temperature was 37 °C, blood pressure = 110/70 mmHg, heart rate = 78 beats per minute, respiratory rate = 30 breaths per minute, and the oxygen saturation was 98% in ambient air. She had pale skin and mucosae, without peripheral edema, no palpable pulse in either pedal artery, but present in tibialis posterior arteries bilaterally. Widespread tenderness of the abdomen without peritoneal irritation signs was discovered upon palpation. The remainder of the examination was normal.

The blood tests showed normal levels of myocardial enzymes, no signs of hepatic or renal involvement, normal levels of serum ions. However, the blood count revealed marked thrombocytosis (709,000/mm^3^), mild leukocytosis (10,750/mm^3^), and normal red blood cell count, with a hemoglobin value of 14.9 g/dL, consistent with the patient’s previous tests. Abdominal CT showed a circumferential parietal thrombosis involving the superior mesenteric artery, originating at the aortic emergence and extending over a length of approximately 12 mm, maintaining a filiform, patent distal lumen with normal filling, both of the artery and the bowel loops (Figure 2).

Based on the clinical signs, blood results and imagistic findings, the positive diagnosis was an exacerbation of chronic mesenteric ischemia. For the motivation of the diagnosis, we considered the following criteria: the presence of postprandial pain, systemic atherosclerosis, the absence of intestinal infarction/necrosis, the lack of metabolic acidosis, and the thrombotic origin of the ischemic event revealed on the abdominal CT [18,19].

Abdominal angiography was performed (Figure 3 left). Selective catheterization of the superior mesenteric artery demonstrated a 12 mm long thrombus, located immediately beyond the aortic emergence, thus reducing the arterial lumen down to 10% of its usual size.

The bone marrow biopsy performed due to a suspicion of ET revealed megakaryocytic hyperplasia. Genetic testing for JAK-2 mutation revealed the presence of the mutation, supporting the diagnosis of ET [20,21].

An interdisciplinary consult with general and vascular surgery concluded that, due to her age and comorbidities, a surgical strategy would impose too much of a risk. She consented to a percutaneous therapeutic procedure.

Endovascular procedure was performed under local anesthesia. Catheterization for percutaneous transluminal balloon angioplasty (PTA) and stent placement was performed via the brachial route. A 0.014-inch guidewire (Terumo, Somerset, NJ, USA) was used to cross the lesion transluminally. A short5F PIER (Cordis, Miami, FL, USA) catheter was used to reach the aortic lumen. A short 6 mm balloon-expandable Cobalt-Chromium stent (Express, Boston Scientific, Natick, MA, USA) was placed retrogradely in the superior mesenteric artery origin. There was no dissection and no residual stenosis (Figure 3 right).

Afterwards, she received dual anti platelet therapy with acetylsalicylic acid 75 mg/day (S.C. Terapia S.A., Iasi, Romania) and clopidogrel 75 mg/day (Sanofi Winthrop Industrie, Amilly, France), along with her previously prescribed medication: enalapril 5 mg/day (S.C. Magistra C&C SRL, Constanta, Romania), indapamide 2.5 mg/day (Labormed Pharma S. A., Bucharest, Romania), and simvastatin 40 mg/day (S.C. Terapia S.A, Iasi, Romania). Once the diagnosis of ET was established, the patient also received hydroxyurea 15 mg/kg/day (Bristol Myers Squibb KFT, Budapest, Hungary). The patient’s condition improved after the angioplasty; after discharge, the clinical course continued to be favorable. One month after the procedure, no residual stenosis was detected, and the patency of the stent was confirmed.

## 3. Discussion

Chronic mesenteric ischemia is a rare disease, which can initially be mistaken for acute ischemia, thus setting an unnecessary surgical indication. Moreover, referral to a center for abdominal surgery in a pandemic context, with consequently reduced addressability, could be a logistic challenge [22,23]. A definitive diagnosis, as well as the location and the therapeutic options, can only be determined by angiography [24].

The therapeutic approach to thrombosis-induced mesenteric ischemia follows two main paths, in addition to non-pharmacological management, which involves fluid resuscitation, the correction of electrolyte imbalances and acid-base status, nasogastric decompression, supplemental oxygen, intravenous unfractionated heparin (unless contraindicated), and broad-spectrum antibiotics (intestinal ischemia causes an early loss of mucosal barrier, leading to a high risk of septic complications) [25].

The first route is the surgical one, which should be followed in case of positive clinical or imagistic signs of advanced bowel ischemia or infarction. Positive peritoneal signs generally imply infarction, rather than just ischemia. Open surgical repair techniques include antegrade or retrograde bypass grafting, or direct reimplantation of the occluded vessel, along with the resection of all non-viable tissues [26]. Although, in some cases, median laparotomy may be urgently indicated in order to evaluate the viability and the extent of necrosis in the intestinal ansae, it has been associated with higher morbidity (5 to 30%) when compared to endovascular repair (0–18%) [27].

The second option is endovascular thrombectomy, which may be achieved via percutaneous mechanical aspiration or thrombolysis, which allows percutaneous transluminal angioplasty, with or without stent placement. Mesenteric angioplasty is to be preferred in elder patients and patients with high-risk comorbidities. Unless thrombolysis is contraindicated, the procedure itself has no absolute contraindications, as long as there are no signs of loss of intestinal ansae viability, but it does have relative contraindications pertaining to anatomical variability, such as highly tortuous aortoiliac arteries, small diameter distal vessels, heavily calcified stenosis, or long segment occlusion [27]. Our patient presented favorable mesenteric anatomy and, due the age-related frailty and minimal invasiveness, was assessed and treated with endovascular thrombectomy and stent placement. 

The prognosis in these cases varies based on the type of mesenteric ischemia, the rapidity of mesenteric insult onset (acute or chronic), the mechanism, the delay in seeking medical attention, the extent and severity of the ischemia, and the presence of infarction, along with individual factors such as age and comorbidities. Additionally, the prognosis is less favorable in acute ischemia, and recurrence is not uncommon, irrespective of initial endovascular or open surgical repair. Our patient had a good short-term outcome, explained by the rapid endovascular therapy and the chronic form of MI, even though age-related frailty and medical comorbidities were present.

Additionally, the newly confirmed ET could represent the main risk factor for the exacerbation of chronic mesenteric ischemia in this patient. The therapeutic options for ET vary, depending on whether the patient is classified as high or low risk for thrombosis, and range from careful observation to low-dose acetylsalicylic acid and, in the upper extreme, cytoreductive drugs, as our patient had received. The association of acetylsalicylic acid with hydroxyurea has been established as a first line treatment for patients at high risk for thrombosis [28]. The short-term prognosis is reliant on the occurrence of thrombotic events; however, the long-term prognosis, with proper management, leads to a similar median life expectancy to that of the general population. Transformation to acute myeloblastic leukemia occurs in 0.6–5% of patients with ET, which is comparable to that found in the healthy population [21]. Considering the life expectancy of our patient, possible side effects of cytoreductive therapy notwithstanding, the main objective of the etiologic management of ET was to decrease the risk of thrombotic events.

Furthermore, the recent COVID-19 infection could increase the risk of mesenteric ischemia in our patient [29]. Based on the literature, the causality is difficult to assess, as this association relies mostly on case reports, observational studies, and reviews [29,30,31]. Furthermore, due to the presence of the ACE-R in enterocytes, the COVID-19-related intestinal dysfunction could originate from both endothelial and non-endothelial dysfunction [32,33].

Overall, we may presume that our patient suffered from silent chronic mesenteric ischemia for several years, according to the natural evolution of atherosclerosis, potentially exacerbated by ET and recent COVID-19 infection, both of which represent thrombotic states.

## 4. Conclusions

When facing symptoms of mesenteric ischemia, one must consider obtaining a full biological profile of the patient in order to diagnose any rare conditions that may contribute to the ischemia. The management of mesenteric ischemia in an elderly patient with multiple comorbidities, previous thromboembolic incidents, recently diagnosed ET, and recent pro-coagulant status due to COVID-19, requires an interdisciplinary approach in solving both the exacerbation and the underlying conditions in order to prevent further thrombotic events.

## Figures and Tables

**Figure 1 medicina-58-01147-f001:**
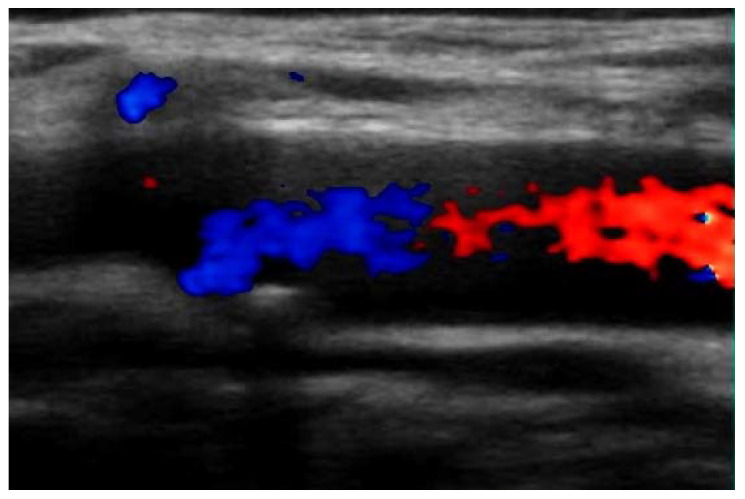
Doppler of the left carotid artery. The image highlights a carotid plaque with consequent flow obstruction (˂50%).

**Figure 2 medicina-58-01147-f002:**
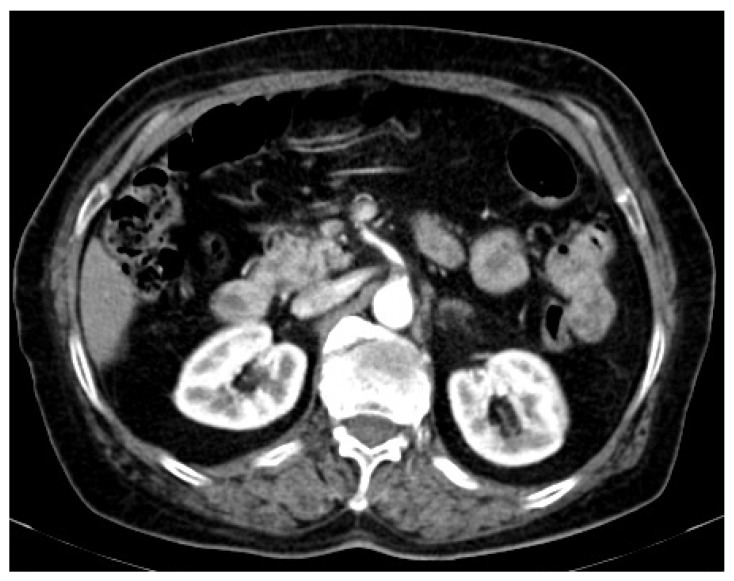
Abdominal CT scan revealing parietal thrombosis of the superior mesenteric artery.

**Figure 3 medicina-58-01147-f003:**
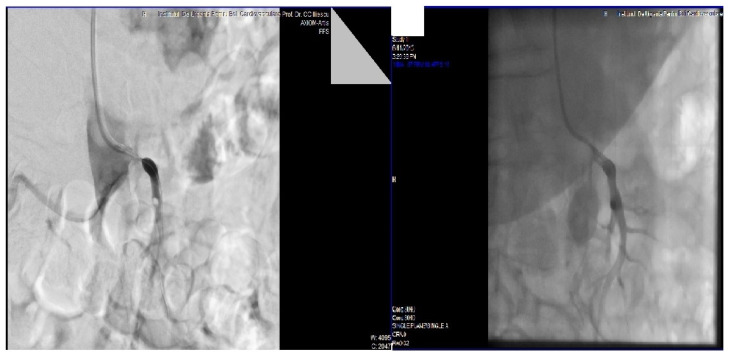
Angiographic image of superior mesenteric artery before (**left**) and after (**right**) stenting.

## Data Availability

Not applicable.

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
