# Peer review of "Mesenteric Ischemia in a Patient with Essential Thrombocythemia: Does COVID-19 Play Any Role? A Case Report and Overview of the Literature"

_medicina, 2022, doi:10.3390/medicina58091147_

Round 1
Reviewer 1 Report
I would recommend supplementing the introduction with more data about COVID-19 and ischemia, COVID-19 and CMI (at least one paragraph).
Considering patient elderliness, are there any relations between advanced age and CMI progression and complications and to be mentioned so far?
Author Response
We thank the reviewer for the useful comments. Please find bellow the point-by-point answers to your questions:
Q1. I would recommend supplementing the introduction with more data about COVID-19 and ischemia, COVID-19, and CMI (at least one paragraph).
R1. We thank the reviewer for the annotation. We added additional information on lines 117-124 and 249-253
Q2. Considering patient elderliness, are there any relations between advanced age and CMI progression and complications and to be mentioned so far?
R2. We are glad for mentioning this issue. We added recent data on this topic. (lines 108-114)

Reviewer 2 Report
This is an interesting manuscript presenting a patient with COVID-19 that had mesenteric ischemia and essential thrombocytopenia. Here are some comments that could be taken into consideration:
1. Line 108: Change COVID to COVID-19
2. The introduction is quite extended relatively to the case presentation and the discussion. It is not necessarily wrong, however, the introduction refers only to the mesenteric ischemia. I would like to see a paragraph referring to COVID-19 and its association with a prothrombotic state. There are several articles approaching this association in the literature, so the authors could develop the introduction towards that direction
3. Similarly, I would like to see a couple of sentences in the introduction section on essential thrombocythemia
4. The figure quality can be significantly improved if the technical details, letters and measuring signs are removed
5. Informed consent statement is mentioned twice although in two slightly different sentences
6. The main issue with this manuscript is that the title implies that the thrombosis was directly associated with COVID-19, but the introduction, the discussion, and, obviously, the results section barely mention COVID-19. Since COVID-19 was diagnosed three months ago, it is hard to believe there is a causal association between COVID-19 and the thrombosis that developed afterwards. Thus, the title is probably misleading. If the authors believe that there is indeed a causal association, more information regarding COVID-19 would be required in all sections of the manuscript. In that case, it would be necessary to include information regarding the episode of COVID-19. Was the patient hospitalized? Was she treated with antivirals? Did she receive any anticoagulants at that time? In that case, it is also important to try to explain through the literature how such a distal diagnosis of COVID-19 could be associated with the current thrombosis
Author Response
Response to reviewer 2
We thank the reviewer for the useful remarks. Please find bellow the point-by-point answer to your comments
Q1. Line 108: Change COVID to COVID-19
R1. We thank the reviewer for the remark. We changed accordantly.
Q2. The introduction is quite extended relatively to the case presentation and the discussion. It is not necessarily wrong, however, the introduction refers only to the mesenteric ischemia. I would like to see a paragraph referring to COVID-19 and its association with a prothrombotic state. There are several articles approaching this association in the literature, so the authors could develop the introduction towards that direction
R2. We are grateful for your comment. We added accordantly additional information on this topic: lines 117-124 and 249-253
Q3. Similarly, I would like to see a couple of sentences in the introduction section on essential thrombocythemia
R3. Thank you for addressing this issue. We added supplementary information regarding ET in lines 125-130
Q4. The figure quality can be significantly improved if the technical details, letters and measuring signs are removed
R4. We thank the reviewer for the comment. We changed them accordantly.
Q5. Informed consent statement is mentioned twice although in two slightly different sentences
R5. We apologies for overlooking this issue. Repeated statement was erased.
Q6. The main issue with this manuscript is that the title implies that the thrombosis was directly associated with COVID-19, but the introduction, the discussion, and, obviously, the results section barely mention COVID-19. Since COVID-19 was diagnosed three months ago, it is hard to believe there is a causal association between COVID-19 and the thrombosis that developed afterwards. Thus, the title is probably misleading. If the authors believe that there is indeed a causal association, more information regarding COVID-19 would be required in all sections of the manuscript. In that case, it would be necessary to include information regarding the episode of COVID-19. Was the patient hospitalized? Was she treated with antivirals? Did she receive any anticoagulants at that time? In that case, it is also important to try to explain through the literature how such a distal diagnosis of COVID-19 could be associated with the current thrombosis
R6. According to the literature, it is difficult to establish an exact causality, considering that some ischemic events were reported even at one year post COVID (Xie et al- a study with over 150000 patients, ref.11). Moreover, the relationship between COVID infection and ischemia has a physiophathologic underlying mechanism (ref 12-14) and is based, manly, on case-reports and reviews (so no direct causality could be proven until now). These information were added into the new manuscript accordantly (see Introduction 117-124 and Disscussion lines 249-253 ).
We considered that ET and COVID-_19 infection represents risk factors rather than etiologic agents, hence changing the title to suggest a less categorical association.

Reviewer 3 Report
Dear authors
Thank you very much for submitting your paper to our journal. Aim of the study is reporting the rare case of acute MI due to the thrombosis related with COVID-19 infection and ET. The cause of acute MI and treatment methods are very interesting. I have some comments on your manuscript.
Major comment
1. Introduction
a. The definition of “Essential thrombocythemia” should be described under the introduction part (may be described these under new paragraph after the description of MI)
2. Case presentation > line 119
a. “Minor form of COVID-19” should be more described about the severity of disease and the definition of “minor form of COVID-19”
3. The most likely diagnosis from the clinical presentation and CTA should be described under case presentation topic. (Example: Acute thrombosis mesenteric ischemia)
4. Line 148 > “Catheterization for PTA “
a. PTA need to write in full letter (percutaneous transluminal balloon angioplasty ?) before the acronym use.
5. Line 149
a. The type/size of introducer sheath, wire, catheter usage and technique to selected to the SMA and technique for pass the lesion with precaution which should be concern should be described.
6. Line 150-151
a. Type of stent should be described (Example: balloon expandable stent or self expandable stent, stent graft or bare metal stent, nitinol or cobalt chromium stent or other material stent ?)
7. Discussion > Line 167-168 > The statement “Chronic mesenteric ischemia is a rare disease, which can initially be mistaken for acute ischemia, thus setting an unnecessary surgical indication”
a. The method of differential diagnosis between acute and chronic MI should be described in this paragraph.
8. Line 177-193 : The treatment of these paragraphs should be specific for acute mesenteric thrombosis to avoid the confusion for reader (Specific treatment of acute emboli is open embolectomy and for thrombosis, treatment of choice is bypass surgery)
9. Line 185 and 193: “embolectomy” should be revised to “thrombectomy” (The term “embolectomy” use for embolic cause of acute ischemia)
10. Line 119-121 The statement “The patient presented severe postprandial pain in the lower abdomen, left flank and left hypochondrium. The pain had appeared a week earlier and progressively worsened, especially after meals.”
a. This sentence means “acute MI” but not clarify about the underlying of chronic MI in this patient (chronic symptoms > 2 week)
11. Line 124-126 The statement “She had pale skin and mucosae, without peripheral edema, no palpable pulse in either pedal artery, but present in tibialis posterior arteries bilaterally. “
a. This statement means the patients had underlying with lower extremity PAD due to atherosclerosis. If atherosclerosis was involved lower extremity, why the posterior tibial arteries were not absent/fainting pulse?
12. Because patients had history of multifaceted involvement (CAD, HTN, Carotid stenosis). So, the evidence of atherosclerosis of other arteries from CTA result should be described under case presentation topic. (CT can provide the information of present or absent of systemic atherosclerosis)
Best wishes
Reviewer
Author Response
We thank the reviewer for the helpful comments. Please find below the point-by-point responses to the issues addressed.
- Introduction
- The definition of “Essential thrombocythemia” should be described under the introduction part (may be described these under a new paragraph after the description of MI)
- We thank the reviewer for the useful comment. We added a supplementary paragraph addressing this topic (lines 125-130)
- Case presentation > line 119
- “Minor form of COVID-19” should be more described the severity of the disease and the definition of “minor form of COVID-19”
- We appreciate addressing this topic. We added additional information on the COVID-19 episode (lines 139-142) and a supplementary ref. (ref 17), accordantly.
- The most likely diagnosis from the clinical presentation and CTA should be described under case presentation topic. (Example: Acute thrombosis mesenteric ischemia)
- We apologize for this omission. We added an extra paragraph about the most likely diagnostic: lines 169-173 Lines 177-182, added as exemplified below as exacerbation of chronic mesenteric ischemia. Please consider our explanations regarding comment 10a
- Line 148 > “Catheterization for PTA “
- PTA need to write in full letter (percutaneous transluminal balloon angioplasty ?) before the acronym use.
- Please accept our apologies for this omission, we have added the full term.
- Line 149
- The type/size of introducer sheath, wire, catheter usage and technique to selected to the SMA and technique for pass the lesion with precaution which should be concern should be described.
- We thank you for this comment. Please find the information on lines 183-188
- Line 150-151
- Type of stent should be described (Example: balloon expandable stent or self expandable stent, stent graft or bare metal stent, nitinol or cobalt chromium stent or other material stent ?)
- Please consider the newly added information: lines 185-188
- Discussion > Line 167-168 > The statement “Chronic mesenteric ischemia is a rare disease, which can initially be mistaken for acute ischemia, thus setting an unnecessary surgical indication”
- The method of differential diagnosis between acute and chronic MI should be described in this paragraph.
- Thank you for the recommendation! We have included what lead us to suspect the pathology as being chronic, rather than acute. (lines 169-173). Two additional references were added to support our statement (ref. 18-19)
- Line 177-193: The treatment of these paragraphs should be specific for acute mesenteric thrombosis to avoid the confusion for reader (Specific treatment of acute emboli is open embolectomy and for thrombosis, treatment of choice is bypass surgery)
- Thank you for the suggestion! Paragraphs were rephrased to improve clarity. Moreover, we emphasized the therapeutic options for thrombotic-induced mesenteric ischemia, as considered relevant to our case.
- Line 185 and 193: “embolectomy” should be revised to “thrombectomy” (The term “embolectomy” use for embolic cause of acute ischemia)
- Thank you for pointing out the oversight! The text has been modified accordingly.
- Line 119-121 The statement “The patient presented severe postprandial pain in the lower abdomen, left flank and left hypochondrium. The pain had appeared a week earlier and progressively worsened, especially after meals.”
- This sentence means “acute MI” but not clarify about the underlying of chronic MI in this patient (chronic symptoms > 2 week)
- We thank the reviewer for the valuable comment. In fact, the lack of a clear consensus with regard to the various forms of mesenteric ischemia according to the symptoms onset leads to intense debates in the literature. We considered the mesenteric ischemia episode of our patient to be rather chronic as a result of various factors: the presence of postprandial pain and the lack of pain "out of proportion" which could have been indicated rather an acute form), the absence of atrial fibrillation (which could be the main risk factor for an acute mesenteric ischemia episode of embolic origin- the most common form), the presence of systemic atherosclerosis (from the patient’s history and newly added Fig.1)- which represents an important risk factor for a chronic form of mesenteric ischemia, the absence of intestinal infarction/necrosis (which usually occurs in acute forms), the lack of metabolic acidosis (which is mostly present in acute forms), the presence on CTA of a circumferential parietal thrombosis (which represents the main cause of chronic mesenteric ischemia). The aforementioned characteristics could be found in various medical databases like uptodate or various guidelines like "ESTES guidelines: acute mesenteric ischaemia -2016"
The symptoms’ onset is indeed debatable: the patient declared that the postprandial pain began 1 week prior to admission. This is, however, subjective and could be insufficient for a categorical positive diagnosis of acute mesenteric ischemia. At your suggestion we reconsidered, however, the diagnosis, as exacerbation of chronic mesenteric ischemia. We added 2 supplementary references, accordingly.
The sentence “Chronic mesenteric ischemia is a rare disease, which can initially be mistaken for acute ischemia, thus setting an unnecessary surgical indication” (discussion section), offers an explanation of how misleading a chronic mesenteric ischemia could be, like in the case of our patient, who suffered from chronic mesenteric ischemia but with a rather acute onset of the pain.
We added, for more clarity, a new paragraph with further explanations on this topic and added the positive diagnostic of exacerbation of chronic mesentheric ischemia (lines 69-173)
- Line 124-126 The statement “She had pale skin and mucosae, without peripheral edema, no palpable pulse in either pedal artery, but present in tibialis posterior arteries bilaterally. “
- This statement means the patients had underlying with lower extremity PAD due to atherosclerosis. If atherosclerosis was involved lower extremity, why the posterior tibial arteries were not absent/fainting pulse?
- Unfortunately, pulse palpation is a subjective maneuver, meaning that anatomical variants or small oversights from the clinician could lead to false negative results. Alternatively, Dieter et al. mention that dorsalis pedis pulse is congenitally absent in 1 in 10 individuals, whereas fewer than 1 in 100 lack tibialis posterior pulse; moreover, they mention that the absence of dorsalis pedis pulse carries low specificity and sensitivity [doi: 10.1002/clc.4950250103]
- Because patients had history of multifaceted involvement (CAD, HTN, Carotid stenosis). So, the evidence of atherosclerosis of other arteries from CTA result should be described under case presentation topic. (CT can provide information of the present or absence of systemic atherosclerosis)
- The abdominal CT revealed a relevant atherosclerotic process of the SMA. In order to prove the existence of systemic atherosclerosis, we added, however, a new image of Doppler ultrasonography image revealing stenosis of the left carotid artery.

Round 2
Reviewer 2 Report
The manuscript has been improved during the revision process.